# The improved and the unimproved: Factors influencing sanitation and diarrhoea in a peri-urban settlement of Lusaka, Zambia

Sikopo Nyambe[1], Lina Agestika[1], Taro Yamauchi[2,3]*

1 Laboratory of Human Ecology, Graduate School of Health Sciences, Hokkaido University, Sapporo, Hokkaido, Japan, 2 Laboratory of Human Ecology, Faculty of Health Sciences, Hokkaido University, Sapporo, Hokkaido, Japan, 3 Research Institute for Humanity and Nature, Kyoto, Japan

* taroy@med.hokudai.ac.jp

**Data Availability Statement:** All Peri-urban sociodemographic & WASH characteristics (Sep-Oct 2018, Lusaka, Zambia) files are available from

## Abstract

Accounting for peri-urban sanitation poses a unique challenge due to its high density, unplanned stature, with limited space and funding for conventional sanitation instalment. To better understand users, needs and inform peri-urban sanitation policy, our study used multivariate stepwise logistic regression to assess the factors associated with use of improved (toilet) and unimproved (chamber) sanitation facilities among peri-urban residents. We analysed data from 205 household heads in 1 peri-urban settlement of Lusaka, Zambia on socio-demographics (economic status, education level, marital status, etc.), household sanitation characteristics (toilet facility, ownership and management) and household diarrhoea prevalence. Household water, sanitation and hygiene (WASH) facilities were assessed based on WHO-UNICEF criteria. Of particular interest was the simultaneous use of toilet facilities and chambers, an alternative form of unimproved sanitation with focus towards all-in-one suitable alternatives. Findings revealed that having a regular income, private toilet facility, improved drinking water and handwashing facility were all positively correlated to having an improved toilet facility. Interestingly, both improved toilets and chambers indicated increased odds for diarrhoea prevalence. Odds of chamber usage were also higher for females and users of unimproved toilet facilities. Moreover, when toilets were owned by residents, and hygiene was managed externally, use of chambers was more likely. Findings finally revealed higher diarrhoea prevalence for toilets with more users. Results highlight the need for a holistic, simultaneous approach to WASH for overall success in sanitation. To better access and increase peri-urban sanitation, this study recommends a separate sanitation ladder for high density areas which considers improved private and shared facilities, toilet management and all-inclusive usage (cancelling unimproved alternatives). It further calls for financial plans supporting urban poor access to basic sanitation and increased education on toilet facility models, hygiene, management and risk to help with choice and proper facility use to maximize toilet use benefit.

the Open ICPSR database. DOI: http://doi.org/10.3886/E117961V1.

**Funding:** Our research activities were supported by: (i) "The Sanitation Value Chain: Designing Sanitation Systems as Eco-Community Value System" Project, Research Institute for Humanity and Nature (RIHN, Project No.14200107). URL: http://www.chikyu.ac.jp/rihn_e/; and (ii) Japan Society for the Promotion of Science KAKENHI to TY. URL: https://www.jsps.go.jp/index.html SN, LA and TY are all members of the RIHN Project. The funders had no role in study design, data collection and analysis, decision to publish, or preparation of the manuscript.

**Competing interests:** The authors have declared that no competing interests exist.

## Introduction

Sustainable Development Goal (SDG) 6 focuses on universal access to improved drinking water and sanitation by the year 2030. Access to basic services such as water, sanitation and hygiene (WASH) is still low in high density peri-urban settlements. This primarily results from their being low income unplanned settlements having limited space and municipal provisions [1]. Consequently, residents use a mix of improved and unimproved WASH facilities [2,3].

In the sub-Saharan nation of Zambia, WASH factors have been found to be responsible for 11.4% of all deaths [4]; only 67.7% and 40% of the population have access to improved drinking water and sanitation respectively [5]. In comparison to national statistics, peri-urban figures reveal that approximately 56% and as much as 90% of the peri-urban population lack access to safe water and sanitation facilities respectively [6].

Poor WASH has also been linked to the nations annual cholera outbreaks which usually emanate from rural fishing villages and peri-urban settlements [7]. During the 2017/2018 rain season, an outbreak of cholera emanating from the peri-urban resulted in 5,905 registered suspected cases, the majority of them (91.7%) from Zambia's capital city, Lusaka [8]. Approximately 70% of the city's population are peri-urban residents; the city is home to 37 peri-urban settlements [5].

Household WASH and sociodemographic data in one peri-urban settlement in Lusaka were collected in order to identify factors associated with household access to improved/unimproved WASH and inform future participatory action research among resident children and youth. As the peri-urban has been a common epicentre of diarrheal disease outbreaks, this article focuses on access to peri-urban sanitation. Key points of focus are commonalities, risk factors and plausible intervention areas. Of particular interest in this article is the nature of sanitation facility owned and/or used by the household, and the factors associated with the use of improved and/or unimproved sanitation facilities. Bearing in mind the 2030 target of universal access to basic/improved sanitation [9] rather than co-use between improved and unimproved facilities, the study took a unique assessment of the simultaneous use of improved toilet facilities and unimproved sanitation in the form of chambers: bucket, pan, plastic or other unsealed containers which are collected or disposed daily in toilets, by informal collectors, with solid waste or thrown as flying toilets [3,10]. As a major goal of meeting the SDG targets is the alleviation of disease risk, household diarrhoea prevalence was also assessed.

Objectives of the study were therefore, to: (i) investigate peri-urban sanitation through determining the associations between household socio-demographic and WASH characteristics, and household sanitation facility, chamber use and diarrhoea prevalence; and (ii) narrow down and recommend plausible interventions focused towards attainment of SDG 6 in the peri-urban/ high density areas for the purpose of informing research, policy and WASH institutions.

## Methodology

The study used an exploratory cross-sectional design, with data collected between September and October, 2018. A brief breakdown of research site selection and sampling procedure is given in Fig 1. A questionnaire and observation checklist were used for data collection (see S1 and S2 Appendices), and findings were analysed using multivariate logistical regression. The following is a detailed description of the research process.

### Research site

A previous WASH assessment informed the selection of the research site (i.e., Stage 1 in Fig 1) [10]. The site was also 1 of 2 informal settlements cited as epicentres of the 2017/2018 cholera

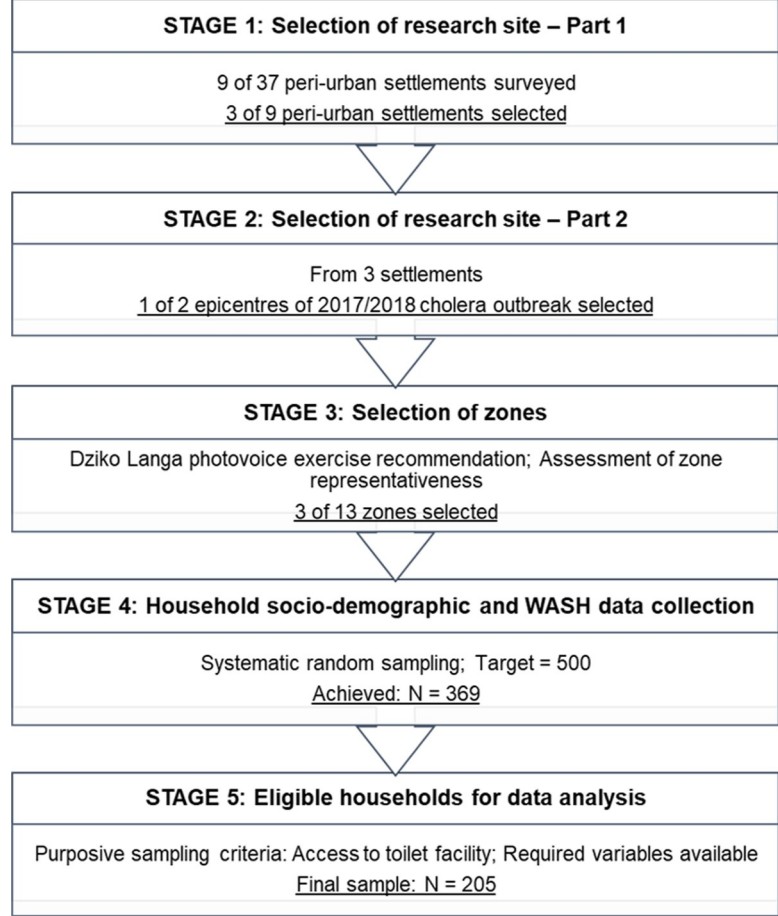

**Fig 1. Flow diagram of sampling procedure.** Caption Credits: Nyambe S, Hayashi K, Zulu J, Yamauchi T. Water, Sanitation, Hygiene, Health and Civic Participation of Children and Youth in Peri-Urban Communities: An Overview of Lusaka, Zambia, Field Research Report 2016. Sanit Value Chain. 2018;Vol. 2(01):39–054, 2018. https://doi.org/10.34416/svc.00010.

outbreak in Lusaka (i.e., Stage 2 in Fig 1) [8]. Within the settlement, 3 out of 13 health zones were selected for data collection (i.e., Stage 3 in Fig 1). The zones were selected in collaboration with a local youth group named Dziko Langa. The groups' decisions were informed by their findings from a photovoice exercise focused on assessing local WASH priorities. Photovoice required participants to take pictures and tell the story of local/peri-urban WASH [11]; the selected zones would also be sites for Dziko Langa's future WASH intervention through action research. Other than recommendations from group members, criteria for zone selection considered availability of WASH facilities, public services and distance from the main road. One of the zones housed the local hospital and several government schools, another housed the biggest market in the settlement, having the 2nd largest number of households among the 13 zones, and the final zone was further in the settlement, off the main road. This variation in development, facilities and population densities among the zones ensured higher possibility of representativeness.

## Sampling and sample size

The number of households in the settlement was 33,185 [10]; the selected zones housed 9,114 households (representing 27.5% of the settlement). Households were selected via systematic

random sampling with data collectors targeting every 5th house and marking each house after data collection to prevent duplication. Sampling commenced from an agreed intersection of the main road/boundary of each zone going into the interior, and zonal boundary markers were clearly defined to all data collectors. In cases where tenants lived in a cluster of houses with their landlords (a common occurrence in Lusaka peri-urban) [12], or where neighbours shared WASH facilities, the 5th household, regardless of who owned the WASH facilities, was the first priority for sampling and the cluster sharing WASH facilities was considered as one household. This was done to avoid duplicating facilities. In several cluster cases, approached households referred data collectors to the landlord, or neighbour in charge of the facility stating the need for permission in order to assess facilities.

The sampling goal was N = 500 for the overall WASH study; a sample size of N = 369 was achieved (i.e., Stage 4 in Fig 1). Purposive sampling was applied on collected data; sampling criteria required households with toilets and information on all required variables (N = 205) (i.e., Stage 5 in Fig 1). Zambia's Fifth National Development Plan indicated that 10% of the peri-urban population had access to 'satisfactory' sanitation facilities [6]. More recent statistics however, indicated that 99% of urban households had access to a facility (regardless of whether it was improved or unimproved as per current study focus) [5]. Using a confidence level of 95% with our sample (N = 205), the latter proportion (99%) gave a confidence interval of ±1.36 while the former (10%) gave a confidence interval of ±4.09.

Sociodemographic data were requested from household heads as they were deemed responsible for and/or knowledgeable on household WASH decision making. The study followed the definition of household head as per the Zambia Living Conditions Monitoring Survey which categorised the household head as the person who normally made daily decisions concerning the running of the household irrespective of gender and/or marital status [5]. Where unavailable, data collectors either collected data from the eldest/responsible available adult if permitted ($\geq$18 years), returned to the household at an alternative time to collect data from the household head directly, or skipped to the next house in the sequence. This was done to ensure that the diversity of household heads in the research area (employed and unemployed) were sampled. In most cases, individuals were not willing to give information without the consent of the household head, as it was the household heads sociodemographic information that was required. In some cases, individuals contacted the household head for permission or to clarify information. The percentage of household head vs. non-household heads who divulged data was 68% vs. 32%.

## Compliance with ethical standards

Prior to the commencement of the study, all processes, documentation and data collection tools underwent ethical screening, and were approved by ERES Converge Ethical Approval Board, Lusaka (Ref. No. 2017-Mar-012) and the Faculty of Health Sciences, Hokkaido University, Japan (Ref. No. 16–103). In line with this, signed informed consent was collected from all participants and all participation was voluntary. Furthermore, data were only collected from persons 18 years and older. The research was conducted under the Sanitation Value Chain Project, registered with the Research Institute for Humanity and Nature based in Kyoto, Japan. The study design, data collection, analysis, article and all other aspects related to the research were fully under the discretion of the researchers.

## Data collection

A questionnaire was used to collect sociodemographic data and household WASH information; questions on socio-demographic data, household sanitation, chamber use and diarrhoea prevalence were extracted for the purpose of the study (see S1 Appendix). Sociodemographic

data was collected in alignment with criteria from the Zambia Demographic and Health Survey 2013–2014 [13]. Since Zambia is a signatory of the SDGs, household sanitation was assessed using the 2017 World Health Organisation and United Nations Children Education Fund Joint Monitoring Programs' Guidelines for WASH (hereinafter referred to as the WHO-UNICEF JMP) [9]. Questions relating to household WASH as per S1 Appendix followed the aforementioned guidelines; a WASH checklist was developed as an observatory guide to determine household WASH service levels (see S2 Appendix).

Both sociodemographic and WASH data were collected using Open Data Kit (ODK) Collect as the phone application for initial data collection and KoBoToolbox as the online data server post-collection. Data collectors had 4 days training on how to use ODK Collect, and fill in the questionnaire and checklist. Note that data collectors entered participant responses in the application, which they later verified for error before upload to the online server. To reduce error, the researcher and research assistants shadowed different pairs of data collectors through the first half of the data collection period.

**Household demographic and WASH questionnaire.**   Sociodemographic data collected from the household head were age, gender, marital status, education level, employment status, income, house ownership and number of household members. For a more in-depth look into peri-urban sanitation, questions were also asked on toilet ownership and management (cleaning and cleaning frequency, maintenance, hygiene). This would help determine internal and external matters of access, control and management of household sanitation.

Data were also collected on the use of chambers and diarrhoea prevalence. Use of chambers is a relatively well known practice in the peri-urban irrespective of an individual's access to sanitation [10]. According to an update of the WHO-UNICEF JMP, chambers fall in the category of unimproved sanitation as they present significant health risks. When disposed in the open or with solid waste, they equate to open defecation [3]. With their normalcy, an analysis of chamber use could show chamber impacts and expose barriers to toilet use in the peri-urban. Lastly, household diarrhoea prevalence was assessed as per previous studies: any household member having 3 or more watery stools within 24 hours in the last 2 weeks [14–16]. This information was also collected to gauge the relationship between peri-urban health (diarrhoea prevalence) and sanitation.

**Household WASH checklist.**   WASH data were collected by viewing the households' water source, sanitation facility, faecal disposal site (e.g., septic tank) and handwashing station/location; where permitted, photographs of WASH facilities were also taken to assist later validation. GPS coordinates of all participating households were also taken for this purpose. Observations facilitated household WASH assessment via the 2017 WHO-UNICEF JMP which categorises WASH facilities into improved (safely managed, basic and limited) vs. unimproved (unimproved and surface water/open defecation) for drinking water and sanitation, and facility (facility with soap and water, and facility without soap and/or water) vs. no facility for hygiene, i.e., handwashing [9].

Households with access to piped water, boreholes or tube wells, protected dug wells, protected springs, and packaged or delivered water sources were categorised as having 'Improved' drinking water. 'Unimproved' drinking water was indicated by households that accessed water from unprotected sources (dug well or spring) and surface water (directly from a river, dam, lake, pond, stream, canal or irrigation canal). Having a handwashing facility, regardless of soap and/or water availability was categorised as 'Facility', whilst the absence of such facilities was categorised as 'No facility'.

Of primary importance to this research was the categorisation of sanitation. Improved facility status was granted to households that accessed flush/pour flush to piped sewer systems, septic tanks or pit latrines, ventilated improved pit latrines, composting toilets or pit latrines with

slabs. 'Unimproved' facility was used to categorise households using pit latrines without slab or platform, bucket latrines, and disposal of faeces in fields, forests, bushes, open bodies of water or other open spaces, or with solid waste [9]. In cases where households had more than one toilet or type of sanitation, the most used by the household was the one assessed.

### Data analysis

Data were analysed using JMP® Pro, Version 13.1.0 (SAS Institute Inc., Cary, NC, 2016) for Microsoft Windows 10 Pro. Descriptive statistics were used to analyse socio-demographic and household WASH characteristics. The association between household heads socio-demographic details and household WASH characteristics was evaluated using multivariate stepwise logistic regression in order to identify a parsimonious set of predictors of toilet facility category, chamber use and diarrhoea prevalence.

To select variables for stepwise regression, bivariate odds ratios were computed between each dependent and independent variable; only those resulting in $p < 0.25$ were included in the multivariate model. For toilet facility, eligible dependent factors were employment, income, toilet ownership, private vs. shared facility, number of households using the toilet, toilet cleaning frequency, drinking water, handwashing, chamber use and diarrhoea prevalence. For chamber use, eligible dependent factors were gender, number of household members, toilet ownership, number of households using the toilet, toilet cleaning responsibility, toilet hygiene, toilet facility and diarrhoea prevalence. Lastly, for diarrhoea prevalence, eligible dependant variables were gender, education, private vs. shared facility, number of households using the toilet, number of persons using the toilet, toilet cleaning responsibility, toilet cleaning frequency, toilet hygiene, toilet facility and chamber use. S3 Appendix shows results of bivariate odds ratios for each independent variable.

As per the Akaike Information Criterion, eligible factors were then computed via a backwards stepwise method to determine factors that significantly contributed to sanitation facility (Improved vs. Unimproved), chamber use (Yes vs. No) and diarrhoea prevalence (Yes vs. No). The p-value threshold for entry and removal into the model to determine adjusted odds ratio was 0.25 and 0.1 respectively. The level of significance was set at $p < 0.05$ with a confidence interval of 95%.

## Results

### Sociodemographic characteristics

Participant sociodemographic characteristics are shown in Table 1. Whilst participant percentages were almost evenly divided by age group, education level, employment status and those owning or renting their residence, the majority were female (83.4%), married/living together (70.7%), receiving irregular income (74.6%) and housing a maximum of 5 persons in their households (62.4%). Based on the varying means and sources of income, several respondents were not able to state a specific or average amount of money they earned per month, so respondents were instead categorised as having regular (known average amount) and irregular (unknown average amount) income. Categorisation of regular income was irrespective of amount, and focused on respondents who could state a known consistent income pattern.

### Household WASH characteristics and diarrhoea prevalence

Table 2 outlines information on the households' WASH status, sanitation characteristics and diarrhoea prevalence. The distribution of characteristics among persons using the toilet, toilet cleaning and hygiene responsibility, and handwashing facility status was relatively even. Majority of toilets were not owned by the household (resident), but externally (74.1%) which

**Table 1. Socio-demographic characteristics of the household head (N = 205).**

| Characteristic | N (%) |
|---|---|
| Age | |
| 18-29yo | 54 (26.3) |
| 30's | 58 (28.3) |
| 40's | 48 (23.4) |
| ≥50 | 45 (22.0) |
| Gender | |
| Male | 34 (16.6) |
| Female | 171 (83.4) |
| Marital Status | |
| Married/Living together | 145 (70.7) |
| Single | 60 (29.3) |
| Education | |
| Secondary/above | 100 (48.8) |
| Primary/below | 105 (51.2) |
| Employment | |
| Employed | 86 (42.0) |
| Unemployed | 119 (58.0) |
| Income | |
| Regular | 52 (25.4) |
| Irregular | 153 (74.6) |
| House Ownership | |
| Resident/Family | 91 (44.4) |
| Rental | 114 (55.6) |
| Household Members | |
| ≤5 persons | 128 (62.4) |
| ≥6 persons | 77 (37.6) |

was also reflected in 80.5% of toilets being shared. The majority of shared toilets were used by ≤5 households (73.2%); the maximum number of households registered as using one toilet was 20 (Median = 3), and the maximum number of persons using one toilet was 33 (Median = 9.5). To incorporate the aspect of toilet sharing into number of toilet users, we considered the sharing of 1 toilet by 2 average households (N = 9.4 persons). As such, toilet users were divided into ≤9 persons (49.3%) vs. ≥10 persons (50.7%).

With multiple users and owners of sanitation facilities, the responsibilities of toilet cleaning, maintenance (in case of toilet damage, or emptying) and hygiene (the supply of hygiene materials such as toilet paper, cleaning materials and handwashing station for example) were divided into resident and external [12]. Resident management of toilet cleaning and hygiene was at 49.8% and 53.7% respectively. Most participants reported that toilet cleaning was done several times a day to daily (92.7%). The majority of toilets (89.8%) underwent a form of maintenance when damaged, malfunctioning or full (including emptying for pit latrines); of the sample, 29.6% of participants attested to use of a chamber. Access to improved toilet facility was at 72.7%, and drinking water at 84.9%. Having a handwashing facility was at 41.0%. Household diarrhoea prevalence within the past 2 weeks was at 8.3%.

## Factors contributing to improved toilet facility access

Table 3 gives results for logistic regression analysis of factors associated with households having access to an improved toilet facility. The significant independent predictors that increased

**Table 2. Household WASH characteristics and diarrhoea prevalence.**

| Characteristic | N (%) |
|---|---|
| Toilet ownership | |
| Resident | 53 (25.9) |
| External (Landlord/Other) | 152 (74.1) |
| Private vs. Shared toilet | |
| Private | 40 (19.5) |
| Shared | 165 (80.5) |
| Households using toilet | |
| ≤5 households | 150 (73.2) |
| ≥6 households | 55 (26.8) |
| Persons using toilet | |
| ≤9 persons | 100 (48.8) |
| ≥10 persons | 105 (51.2) |
| Responsible: Toilet cleaning | |
| Resident | 102 (49.8) |
| External (Landlord/Other) | 103 (50.2) |
| Toilet cleaning frequency | |
| Several times a day to Daily | 190 (92.7) |
| Several times a week to Never | 15 (7.3) |
| Toilet maintenance (+Emptying) | |
| Yes | 184 (89.8) |
| No | 21 (10.2) |
| Responsible: Toilet Hygiene | |
| Resident | 110 (53.7) |
| External (Landlord/Other) | 95 (46.3) |
| Toilet facility | |
| Improved | 149 (72.7) |
| Unimproved | 56 (27.3) |
| Drinking water | |
| Improved | 174 (84.9) |
| Unimproved | 31 (15.1) |
| Handwashing | |
| Facility | 84 (41.0) |
| No Facility | 121 (59.0) |
| Chamber use | |
| Yes | 61 (29.8) |
| No | 144 (70.2) |
| Diarrhoea prevalence | |
| Yes | 17 (8.3) |
| No | 188 (91.7) |

odds for improved toilet facility access and use were regular household income (AOR = 6.29, 95% confidence interval [CI]: 1.71–23.14), having a private toilet (AOR = 4.43, 95% CI: 1.42–13.87), having access to a handwashing facility (AOR = 7.98, 95% CI: 2.90–21.95), improved drinking water (AOR = 4.80, 95% CI: 1.68–13.77), and households with diarrhoea prevalence (AOR = 10.89, 95% CI: 1.54–77.10). The odds of having access to an improved toilet facility were less for persons who used chambers (AOR = 0.27, 95% CI: 0.12–0.64).

**Table 3. Logistic regression analysis of factors associated with improved toilet facility access.**

| Variable | Improved facility, N (%) | AOR (95% CI) |
|---|---|---|
| Income | | |
| Regular | 49 (94.23) | 6.29 (1.71–23.14)** |
| Irregular | 100 (65.36) | 1 |
| Private vs. Shared toilet | | |
| Private | 34 (85.00) | 4.43 (1.42–13.87)* |
| Shared | 115 (69.70) | 1 |
| Handwashing | | |
| Facility | 78 (92.86) | 7.98 (2.90–21.95)** |
| No Facility | 71 (58.68) | 1 |
| Drinking water | | |
| Improved | 139 (79.89) | 4.80 (1.68–13.77)** |
| Unimproved | 10 (32.26) | 1 |
| Chamber use | | |
| Yes | 36 (59.02) | 0.27 (0.12–0.64)** |
| No | 113 (78.47) | 1 |
| Diarrhoea prevalence | | |
| Yes | 15 (88.24) | 10.89 (1.54–77.10)* |
| No | 134 (71.28) | 1 |

*$P < .05$;

**$P < .01$

## Factors contributing to chamber use

Table 4 shows the logistic regression analysis of factors associated with using a chamber. Independent predictors of using a chamber were being female (AOR = 3.41, 95% CI: 1.10–10.53),

**Table 4. Logistic regression analysis of factors associated with chamber use.**

| Characteristics | Using a chamber, N (%) | AOR (95% CI) |
|---|---|---|
| Gender | | |
| Male | 6 (17.65) | 1 |
| Female | 55 (32.16) | 3.41 (1.10–10.53)* |
| Toilet ownership | | |
| External (Landlord/Other) | 25 (47.17) | 1 |
| Resident | 36 (23.68) | 4.14 (1.81–9.48)** |
| Responsible: Toilet hygiene | | |
| Resident | 24 (21.82) | 1 |
| External (Landlord/Other) | 37 (38.95) | 3.36 (1.56–7.25)** |
| Diarrhoea prevalence | | |
| No | 50 (26.60) | 1 |
| Yes | 11 (64.71) | 6.49 (1.99–21.11)** |
| Toilet facility | | |
| Improved | 36 (24.16) | 1 |
| Unimproved | 25 (44.64) | 2.33 (1.12–4.87)* |

*$P < .05$;

**$P < .01$

residents ownership of the toilet (AOR = 4.14, 95% CI: 1.81–9.48), and toilet hygiene being handled externally (AOR = 3.36, 95% CI: 1.56–7.25). Additionally, chamber users had higher odds of having diarrhoea (AOR = 6.49, 95% CI: 1.99–21.11) and were more likely to have an unimproved toilet facility (AOR = 2.33, 95% CI: 1.12–4.87).

Table 4 indicated that the odds of chamber use were higher for households with access to unimproved toilets. Additional data analysis further revealed that unimproved toilets were more likely owned by residents than external toilet owners like landlords (OR = 2.46, 95% CI: 1.26–4.80; p < .01). Moreover, resident/family house ownership also increased the odds of having access to private facilities (OR = 4.38, 95% CI: 2.04–9.39; p < .01)

### Factors contributing to household diarrhoea prevalence

Table 5 shows the logistic regression analysis of factors associated with household member diarrhoea prevalence in the past 2 weeks. Number of households using a toilet and whether a toilet was private or shared did not offer any significant result to diarrhoea prevalence. Higher odds were found however, for having a toilet used by $\geq 10$ people and having diarrhoea (AOR = 3.80, 95% CI: 1.11–13.08). The odds for having diarrhoea were found to be lower for persons not using a chamber (AOR = 0.16, 95% CI: 0.05–0.48) and using an unimproved toilet facility (AOR = 0.18, 95% CI: 0.04–0.90). Access to improved drinking water and having a handwashing facility gave no significant results.

## Discussion

### Sociodemographic characteristics

Participant socio-demographics (see Table 1) revealed some important characteristics to consider about peri-urban residents and lifestyle. Consistent with a previous study [12] but inconsistent with government data [5], female headed households were most common (83.4%); and most respondents were either married or living together (70.7%). The study also had 26% of household heads in the age range of 18–29 years. There could be several reasons for this finding. Firstly, the definition of household head is not linked to age, gender, marital or economic status; primary focus is on normal daily decision making pertaining to running of the household [5]. Secondly, national statistics show that women have higher poverty levels, possibly impacting female residential choices [13]. Thirdly, Zambia has a relatively young population: over 60% are under 25 years of age, with a life expectancy of 49 and 53 years for men and

**Table 5. Logistic regression analysis of factors associated with household diarrhoea prevalence.**

| Characteristics | Having diarrhoea, N (%) | AOR (95% CI) |
|---|---|---|
| Persons using toilet | | |
| $\leq 9$ persons | 4 (4.00) | 1 |
| $\geq 10$ persons | 13 (12.38) | 3.80 (1.11–13.08)* |
| Chamber use | | |
| Yes | 11 (18.03) | 1 |
| No | 6 (4.17) | 0.16 (0.05–0.48)** |
| Toilet facility | | |
| Improved | 15 (10.07) | 1 |
| Unimproved | 2 (3.57) | 0.18 (0.04–0.90)* |

*P < .05;
**P < .01

women respectively [13]. Diseases such as malaria and tuberculosis are some that have impacted the Zambian population pyramid, leaving several young and more elderly persons to fend for even younger family, bearing economic impacts. National statistics show that the largest age group of household heads is 18–29 years [5]. Of the overall sample, 58.0% were unemployed, higher than the 31% registered across the peri-urban [17]. Only 25.4% received regular income. Whilst income level has been noted to have an impact on sanitation [18], the study findings were linked to income consistency.

In addition to the status of the household head, several studies have linked house ownership to WASH decision making [12,19,20] with landlords in most instances, having more say on household WASH than their tenants (residents). The sample offered a good balance between participants who were renting houses (55.6%) and those staying in their own, or family owned households (44.4%). Ownership of the household by the resident, or family meant more autonomy on WASH decision making [12,19,20]. Lastly, being a high density area, the number of household members was considered. According to the 2015 Living Conditions Monitoring Survey, average household size in urban Lusaka is 4.7 persons [5].

## Household WASH characteristics and diarrhoea prevalence

Just as household ownership has an impact on WASH decision making and management, toilet ownership has an impact on sanitation decision making and management (toilet type, cleaning, cleaning frequency, maintenance and hygiene), determining responsible persons. In the peri-urban where shared WASH is a commonality, these could be the resident, neighbour, landlord, family member or a private/public patron [12,19]. The aspect of responsibility seeks to discuss the level of autonomy for household sanitation and the subsequent bureaucracies that arise from having joint responsibility for, having no responsibility for, or being at the mercy of a second party's decision making. This raises questions like: how free do residents feel to use the facilities? To what extent can residents choose or make amendments to their sanitation? How quickly can/do external parties react to sanitation challenges? How much liability is placed on residents? Residents owning their own toilet was only at 25.9%, with 80.5% of toilets being shared by 2 or more households. Toilet sharing is highly characteristic of peri-urban settlements due to insufficient space for toilet construction and land tenure for example [19,20], and has been more recently encouraged by WHO as an acceptable alternative to not owning a toilet in high density areas [1].

Toilet cleaning was at 49.8% for residents vs. 50.2% external; toilet hygiene was the inverse at 53.7% for residents and 46.3% for external persons. Toilets were said to be cleaned at least daily (92.7%). Toilet maintenance, a less frequent need, was not done by 10.2% of the households. Access to improved toilet facility was at 72.7%. Due to the facility focus of the study however, this statistic is not easily comparable with government peri-urban data which includes non-facility sanitation under the unimproved bracket. For peri-urban access to improved drinking water, current study findings were almost 2 times higher than government statistics (84.9% vs. 44%) [5]. This could be due to the location of 2 of the 3 zones where data collection was done (closer to the main road, and public facilities), warranting an easier access to basic services and facilities [21].

Despite all households having access to toilet facilities, use of chambers was at 29.8%. Multivariate stepwise logistic regression computed by this study offered insight towards understanding why chambers still maintained relevance among persons with toilet access, even of improved level. Finally, household diarrhoea prevalence for the last 2 weeks was at 8.3%. Data collection was done in September-October, Zambia's hot and dry season. During this time of year, there is no rainfall and therefore, diarrhoea prevalence is generally low [13,22]. As the

point of understanding diarrhoea prevalence was to understand risk related to sanitation choices, assessing risk during low prevalence periods would give better revelations pertaining to sanitation choices.

## Factors contributing to improved toilet facility access

The household head having regular income increased the odds of having an improved toilet by 6.3% (AOR = 6.29, 95% CI: 1.71–23.14). Several studies have linked sanitation choices to income, economic status and willingness to pay for services amongst others [10,12,20,23]. Despite access to sanitation being declared a basic human right however, it still comes at a cost which several governments and citizenry cannot afford [23]. This finding indicates sanitation as an investment; with regular income supporting planning, peri-urban residents made the effort towards accessing improved toilet facilities. It also supports the possible benefits of subsidies, payment and investment plans in the area of sanitation acquisition [18].

More often, private toilets proved to be improved facilities (AOR = 4.43, 95% CI: 1.42–13.87). With this result, it can be assumed that in addition to regular income, having private facilities gave more autonomy for choice on type of sanitation procurement [20]. With the more recent WHO Guidelines on Sanitation and Health considering shared toilets as a solution in densely populated areas [1] running alongside the popularity of shared facilities as per our sample (80.5%), collaborations between households for the procurement of improved shared toilet facilities might pose as a suitable solution.

In a study by Tidwell et.al. focused on shared facilities, findings indicated that toilet owners (predominantly landlords) worried about their tenants' ability to afford improved sanitation and thus, opted for cheaper toilet models [12]. Their successful intervention towards improvement of peri-urban sanitation facilities through creating dialogue among landlords and their tenants allowed for joint autonomy, collaboration and decision making towards access to improved sanitation. It also opens the door to more communal and social sanitation opportunities.

Households' availability of a handwashing facility also increased the odds of having an improved toilet (AOR = 7.98, 95% CI: 2.90–21.95). Improved toilets were also significantly correlated to having improved drinking water access (AOR = 4.80, 95% CI: 1.68–13.77). Knowledge on handwashing is often revealed through an analysis of WASH knowledge, attitudes and practices, or linked to education [24]. In this study, however, household heads education level bore no significance. Rather, similar to having a handwashing facility, the availability of accessible water for toilet flushing, cleaning, handwashing and hygiene would be a plausible consideration to determine the type of sanitation facility selected by the household [25]. As such, improved water access would preclude greater investment in toilet facility and higher likelihood of access to handwashing facility (both facilities requiring water availability).

A seemingly unexpected result was that having an improved toilet facility increased the odds of household diarrhoea prevalence by 10.9% (AOR = 10.89, 95% CI: 1.54–77.10). It must be stressed at this juncture that diarrhoea, beyond being waterborne, is spread through faecal oral transmission [1]. Toilets are therefore likely places for faecal contamination, particularly when proper toilet structure, maintenance, use and hygiene are not considered. This prompts sanitation recommendations to go beyond encouragement towards procurement of improved toilet facilities to more education on toilet hygiene and maintenance. Blind recommendation towards use of improved toilets minus consideration of these factors may reduce open defecation, but increase toilet users' access to faecal contamination, thereby escalating risk of contamination and diarrhoea prevalence through toilet use [25,26]. It should be noted that though the result was significant (p < .05), the 95% CI range for diarrhoea prevalence was quite wide (95% CI: 1.54–77.10) indicating that though valid, this result may not be a good reflection of this specific sample.

Lastly, access to an improved sanitation facility reduced the odds for chamber use (AOR = 0.27, 95% CI: 0.12–0.64). With most improved facilities being private, having a hand-washing facility and having access to improved drinking water supply, it could be assumed that the level of convenience offered did not warrant the need for alternative sanitation. This is a positive result, indicating the suitability of the sanitation system for peri-urban residents, particularly when all WASH facilities were available and of improved status [25]. It also offers credence to the SDG targets 1.4 [1] relating to the need for universal acquisition of basic services (inclusive of basic WASH).

## Factors contributing to chamber use

All chamber users attested to having access to a toilet. As such, findings show chambers as complementary to the primary toilet facility regardless of whether the toilet was improved (24.16%) or unimproved (44.64%). This chamber use despite access to toilet facilities indicates inefficiencies with the primary toilet facility for users. For successful intervention towards the eradication of open defecation and a complete move to improved sanitation, these inefficiencies must be explored. This requires looking at chambers as a chosen alternative to both open defecation and toilet facilities.

Findings indicated that odds of using a chamber were higher for those having an unimproved facility (AOR = 2.33, 95% CI: 1.12–4.87). Chamber use was also higher when residents owned their own toilet facility (AOR = 4.14, 95% CI: 1.81–9.48). Studies have found that toilet sharing, more common with external toilets, had an impact on freedom of toilet use [23,27]; and as such, residents owning their own toilet would be expected to offer more freedom of toilet use to the household. Whether a toilet was private or shared however, rendered an insignificant result.

A further look indicated unimproved toilets as more likely owned by residents (OR = 2.46, 95% CI: 1.26–4.80) and that resident/family owned houses had increased odds of accessing private facilities (OR = 4.38, 95% CI: 2.04–9.39). With residents already owning private, unimproved toilet facilities, use of chambers would firstly, more likely create minimal tension to users as there would be no major shift in sanitation level (both are unimproved forms of sanitation). Note also, that there are several overlaps in the reasons for open defecation [27] and chamber use as indicated in the current study, i.e., gender restrictions, toilet ownership and hygiene. Secondly, chambers may in some instances, carry more benefit to users in terms of comfort and/or ease of use when compared to their unimproved toilet facility.

That residents would own private toilet facilities in itself indicates the household will to have their own sanitation facilities. As much as results indicated positive correlations between having a private toilet and access to improved facilities, pairing this finding with the cost implications of having an improved toilet (see Table 3) may indicate some opportune benefits for toilet sharing in relation to acquisition of improved toilets amongst the urban poor seeking to own facilities, but limited by cost.

A third possibility could be that residents ownership of their own private facilities averted social pressures for good sanitation practices that may come from the use of shared facilities, i.e., cleaning, maintenance and hygiene [28]. However, this finding was not corroborated with study results. Chamber use was actually more likely when toilet hygiene was handled externally (AOR = 3.36, 95% CI: 1.56–7.25). If responsible persons did not fulfil their duty, toilet users could find it more convenient to use chambers and make use of private hygiene materials; rendering the use of a toilet hygienically insignificant [26,27].

Studies covering toilet hygiene for shared facilities have indicated the challenges of shared facilities in comparison to private ones, citing the importance of duty rotas and accountability

for improved toilet access and use [12]. There was however, no significant result rendered between private vs. shared toilets and toilet hygiene, cleaning, cleaning frequency and maintenance in the present study. Social pressure for the improvement of sanitation has been used by a number of studies successfully [12,29], and could be an avenue worthy of more research for shared facilities in high density areas.

Findings revealed that gender also played a role in chamber use with females having higher odds for use (AOR = 3.41, 95% CI: 1.10–10.53). In line with previous studies [10,26,30], chambers were often considered convenient, private and safe. With pit latrines being outdoor sanitation facilities, use at late hours carried risk, particularly to female toilet users who feared being attacked or harassed by male users. Chambers were also found convenient in times of illness, where constant journeying to the toilet would be strenuous, driving home that the toilet model was not convenient for all toilet users.

Lastly, chamber use increased the odds of household diarrhoea prevalence (AOR = 6.49, 95% CI: 1.99–21.11). This is most likely due to faecal management before and after disposal which creates opportunity for faecal contact [30]. Chambers can be used inside or outside the house. As diarrhoea is spread through faecal contamination, poor storage of faeces within the house increases the risk of ingestion of faeces. With poor storage and usage, spillage, disposal, flies and other house insects, rodents and small children all become actors in increasing faecal contact within and around the household.

If chambers are reusable, cleaning them also poses a health risk through increased faecal contact. If not, chambers can be disposed of in the toilet (depending on the toilet and chamber type, this could lead to blockage and/or failure to empty the facility), with solid waste or thrown as a flying toilet (tossed in an open space) [30]. Disposal into open spaces or solid waste is part of the definition of open defecation [9] which has been proven a health risk increasing diarrhoea prevalence.

## Factors contributing to household diarrhoea prevalence

With both improved and unimproved sanitation bearing risk to household diarrhoea prevalence (see Tables 3–5), further analysis of peri-urban socio-demographics linked to diarrhoea prevalence and sanitation characteristics were made. Interestingly, there was no significant risk between diarrhoea prevalence and drinking water or handwashing. There were also no significant findings linking household diarrhoea prevalence and the frequency of toilet cleaning or if toilet maintenance and emptying was conducted.

The only significant result found in addition to having an improved toilet facility and using a chamber was the number of persons using the toilet. Toilets used by ≥10 persons were found to increase the risk of household diarrhoea prevalence (AOR = 3.80, 95% CI: 1.11–13.08). Rather than households, the focus on number of persons using the toilet allows a more direct count of users, bearing in mind household dynamics, i.e., the extended family system and communal society. It takes into account both the formal and informal nature of toilet sharing which private toilets are not removed from due to the fact that some private toilets may have more usage than shared toilets due to the number of household members and overall users. That said, number of households using a toilet and whether a toilet was private or shared did not offer any significant result to diarrhoea prevalence.

Attention to and control of the number of users may help to tackle aspects of overuse, misuse and subsequent faecal contamination. With the status quo of the peri-urban however, this act may not be feasible: space for toilet construction may be lacking and the costs of management for additional toilet facilities would be considered high [23]. Nevertheless, the finding reiterates firstly, that the call to end open defecation primarily through the use of toilet facilities shifts

faecal contamination points from open air locations to toilets, defeating the purpose of installation and use of these facilities [23,25,26]. Secondly, that in the promotion of toilet ownership and usage, education on how to use and maintain facilities should be considered a package deal to allow the reasons for promoting toilet use against open defecation to retain meaning [23,25,26]. An important point to be garnered from the results is the inability of sanitation facilities on their own, whether improved or unimproved, to alleviate the disease burden. Proper use and maintenance must be considered to allow safe use of facilities by multiple users.

### Limitations of the study

The sufficient yet small sample size would mean that a larger, more spread out sample may grant more detail about the nature of peri-urban sanitation. That said, it is not possible to generalise these findings across all national and international peri-urban settlements. Cross-sectional studies conducted at a different time point may also give more information on household diarrhoea prevalence and its implications on chamber usage. Lastly, tenants' opting out of the study in preference for their landlords' participation may have had an impact on findings.

## Conclusion

Key findings of the study indicate a duality of peri-urban sanitation, with households making use of both improved and unimproved sanitation. Sociodemographic characteristics related to use of improved toilet facility and chambers were income and gender respectively. The impact of income on sanitation is a reflection of the cost implications that hinder the right to sanitation for the urban poor; whilst the gender disparity on chamber use indicates the diverse needs of women and girls, and the subsequent social disparities often overlooked relating to the adequate provision of peri-urban sanitation.

Findings also highlighted an interlinkage between household WASH access and quality, with the ownership of an improved toilet facility predicting improved drinking water, presence of hygiene facility and lowering the odds of chamber use (unimproved sanitation methods), but like chambers, having high odds for household diarrhoea prevalence. This indicates inefficiencies with the system requiring alternatives and a failure of the facility to protect users' from faecal contamination. The result prompts a shift towards education on proper toilet facility use and management to reduce health risk in high density areas, particularly with an increased number of users heightening risk. For unimproved toilet users (the more likely to use chambers), it indicates the ease of use within service level brackets (unimproved facility to unimproved facility). With residents seeking to own private toilets regardless of service level, the quality of the facility owned could be accounted to cost.

In summary, in order to truly meet and achieve the intended benefits of SDG targets towards eradication of open defecation towards improved health and well-being in the peri-urban, the duality of peri-urban sanitation must be addressed. Whilst improved sanitation facilities hold some benefit, the current sanitation systems used in peri-urban Lusaka, Zambia do not fully cater for the needs of the urban poor, women and girls, being inaccessible by cost and, gender and social dynamics respectively.

### Recommendations for peri-urban sanitation

As indicated in the WHO-UNICEF Core Guidelines, sharing of toilets is a plausible solution for high density sanitation. Interventions focused on collaborations between households for the procurement of improved shared toilet facilities would aid in a move towards improved sanitation access [12,31]. Creating collaborations would tackle aspects of improved toilet construction and maintenance for joint, landlord and public toilets. With results indicating a

recognition and willingness by residents to own toilets despite monetary constraints, financial strategies such as pooling of funds and payment plans can be considered/encouraged for the urban poor, aiding towards the procurement and construction of improved private, shared and public toilet facilities. Considering the high cost that current toilet models already have despite their inability to cover all user requirements, greater value would be gained by users for a model that, despite costs, can cover all required needs. More so, when used by neighbourhoods as public facilities, these models could become sources of communal income. Similar systems could also be trialled for communal drinking water and handwashing improvements.

With peri-urban WASH proving to be quite communal (shared facilities) rather than private (per household), a WASH ladder for high density areas might prove beneficial, taking into account facility management, and common cultural, demographic needs and differences. As this study primarily focused on peri-urban sanitation, a High Density Sanitation Ladder (Fig 2) was created for consideration through amending the 2017 WHO-UNICEF JMP sanitation ladder (changes to the original ladder are indicated in bold) [9].

| HIGH DENSITY SERVICE LEVEL | DEFINITION |
|---|---|
| SAFELY MANAGED | Use of improved **private or shared** facilities, **usable by all toilet users, at all times (no co-use of unimproved sanitation) with an available responsibility plan or rota** and where excreta are safely disposed of in situ or transported and treated offsite |
| BASIC | Use of improved **private or shared** facilities, **usable by all toilet users, at all times (no co-use of unimproved sanitation) with an available responsibility plan or rota** |
| LIMITED | Use of improved **private or shared** facilities |
| UNIMPROVED (No change) | Use of pit latrines without a slab or platform, hanging latrines or bucket latrines [Chambers come here] |
| OPEN DEFECATION (No change) | Disposal of human faeces in fields, forests, bushes. Open bodies or water, beaches or other open spaces, or with solid waste [Depending on disposal, chambers come here] |

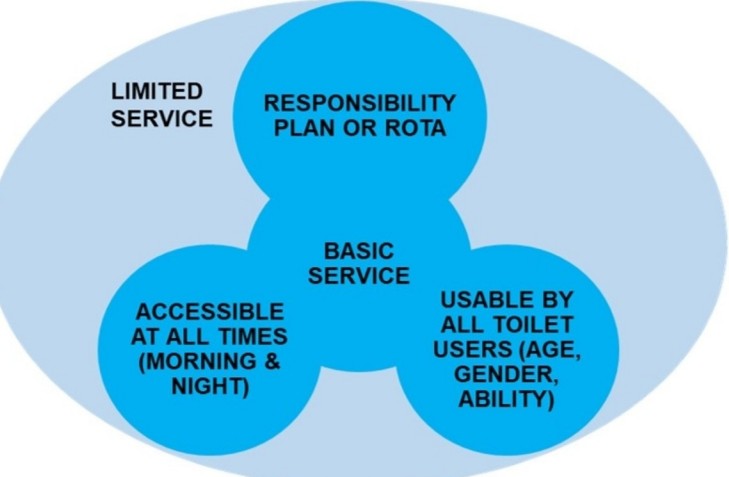

**Fig 2. Recommended high density sanitation ladder.** Caption Credits: Progress on Drinking Water, Sanitation and Hygiene: 2017 Update and SDG Baselines. Geneva: World Health Organization (WHO) and the United Nations Children's Fund (UNICEF), 2017. Licence: CC BY-NC-SA 3.0 IGO.

The ladder incorporates the unique sanitation needs in high density areas through taking note of universal use, complete access and sanitation management regardless of toilets private or shared status. That said, private/shared status has no impact on sanitation level in the suggested model. The upgrade from limited to basic is based on the limited facility being usable by all toilet users, at all times (no co-use of unimproved sanitation) with an available responsibility plan or rota. The upgrade from Basic to Safely Managed contains all these plus faecal disposal as per the original 2017 WHO-UNICEF JMP model. Further studies can be done to look at water and hygiene in high density areas. Additionally, more intervention studies can be done to look into the possible benefits of using social pressure for the improvement of shared sanitation.

Based on the health impacts of chamber use and it's similarities to open defecation, future assessments to determine progress on open defecation should consider all modes of household sanitation including chamber use regardless of households' available sanitation facility. This will help in tackling all forms of unimproved sanitation simultaneously to avoid shifting within sanitation ladder brackets and rather, encourage upgrading.

## Supporting information

**S1 Appendix. Adapted from the 'Household demographic and WASH questionnaire'.** Extracted sections: Sociodemographic data, sanitation, diarrhoea prevalence and chamber use. (PDF)

**S2 Appendix. Water, sanitation and hygiene checklist.** Based on WHO-UNICEF JMP WASH service level criteria. (PDF)

**S3 Appendix. Bivariate odds ratio results.** Independent variables: Toilet facility, chamber use and diarrhoea prevalence. (PDF)

## Acknowledgments

We would like to acknowledge the valuable assistance of Mr. Ian Saungweme and Mr. Emmanuel Mambwe who supported the data collection training and activities. Sincere gratitude is also extended to the Dziko Langa Club and its members for their participation in the data collection exercise, and members of the Lab of Human Ecology, Graduate School of Health Sciences, Hokkaido University for their assistance in verification and scrutiny of data analysis and discussions. Lastly, but not the least, we would like to thank our research partners at the University of Zambia in Lusaka, and our study participants who opened their homes to us for the sake of the research.

## Author Contributions

**Conceptualization:** Sikopo Nyambe.

**Data curation:** Sikopo Nyambe, Lina Agestika.

**Formal analysis:** Sikopo Nyambe.

**Funding acquisition:** Taro Yamauchi.

**Investigation:** Sikopo Nyambe.

**Methodology:** Sikopo Nyambe.

**Project administration:** Taro Yamauchi.

**Supervision:** Taro Yamauchi.

**Validation:** Lina Agestika.

**Visualization:** Sikopo Nyambe, Lina Agestika.

**Writing – original draft:** Sikopo Nyambe.

**Writing – review & editing:** Sikopo Nyambe, Lina Agestika.

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
