## [Decision Letter · Decision Letter 0]

7 Apr 2020

PONE-D-20-05682

The improved and the unimproved: Factors associated with peri-urban sanitation in Lusaka, Zambia

PLOS ONE

Dear Prof. Yamauchi,

Thank you for submitting your manuscript to PLOS ONE. After careful consideration, we feel that it has merit but does not fully meet PLOS ONE’s publication criteria as it currently stands. Therefore, we invite you to submit a revised version of the manuscript that addresses the points raised during the review process.

We would appreciate receiving your revised manuscript by May 21 2020 11:59PM. To enhance the reproducibility of your results, we recommend that if applicable you deposit your laboratory protocols in protocols.io, where a protocol can be assigned its own identifier (DOI) such that it can be cited independently in the future. For instructions see: http://journals.plos.org/plosone/s/submission-guidelines#loc-laboratory-protocols

We look forward to receiving your revised manuscript.

Kind regards,

Khin Thet Wai, MBBS, MPH, MA (Population & Family Planning Resear

Academic Editor

PLOS ONE

Journal Requirements:

2. Please note that the use of the word "slum" throughout the manuscript may be considered controversial and offensive. Please revise the manuscript to omit this word; this term can be substituted with other descriptors you have used such as "low income unplanned settlements" or "peri-urban settlements", which are appropriate.

Additionally, you mention a questionnaire and WASH checklist in the text, but do not include these. Please upload them as supporting files, or alternatively you may include them within the main manuscript or via a link to an external website

Additional Editor Comments (if provided):

This policy-linked study focused the epicenter of cholera outbreak in a periurban slum that comprised informal settlements and followed STROBE Guidelines in reporting the cross-sectional studies with some exceptions.

Unlike several studies, new features included the ascertainment of associated factors for three outcome events: access to sanitary facilities, chamber use and reported prevalence of diarrhoea.

Authors should consider the following points apart from the reviewers' comments to improve the readability and scientific integrity.

1. To be logical, please modify the title as:

"The improved and the unimproved: Factors influencing sanitation and diarrhoea in a periurban slum of Lusaka, Zambia"

2. To check and correct the grammatical errors throughout

and the citation style should follow the submission guidelines.

3. Methodology:

- To add the study design and the study period.

- To add the sample size determination

- To add the flow diagram for depicting the sampling procedure

4. Please consider condensing the conclusion section.

There is a repetition in key findings and interpretation for instance:

LINES : 486-489

Reviewers' comments:

Reviewer's Responses to Questions

**Comments to the Author**

1. Is the manuscript technically sound, and do the data support the conclusions?

Reviewer #1: Partly

Reviewer #2: Yes

2. Has the statistical analysis been performed appropriately and rigorously? 

Reviewer #1: No

Reviewer #2: Yes

3. Have the authors made all data underlying the findings in their manuscript fully available?

Reviewer #1: Yes

Reviewer #2: Yes

4. Is the manuscript presented in an intelligible fashion and written in standard English?

Reviewer #1: Yes

Reviewer #2: Yes

5. Review Comments to the Author

Reviewer #1: Line 92 : In cases where household heads were absent who did you survey?

Line 168 : Add citation for data analysis software used

Data analysis section needs elaboration for a better understanding (How was multivariate stepwise logistic regression computed ? How was your logistic regression models constructed ?

Line 218 : "being a private toilet" It should be "having a private toilet"

Reviewer #2: The manuscript is technically sound and the data supports the conclusions. The statistical analysis has been performed well. All data has been made available according to the PLOS data policy. To a very large extent, the manuscript is presented in an intelligible fashion and written in standard English. The article which is part of a larger study has been well described. It would however be useful if the authors gave a bit more information about the larger study to put the paper in context.

The research site was 1 of 2 informal settlements cited as epicentres of the 2017/2018

cholera outbreak in Lusaka. Within the settlement, 3 health zones were selected for data collection. How many health zones are within the settlement and, what informed the decision to select three health zones?

Kindly cite the ethics clearance certificate number for the 2 institutions as evidence of clearance

In cases where tenants lived in a cluster of houses with their landlords (a common occurrence in Lusaka peri-urban) only one household was targeted. What criteria was used to decide which household to target.

Both sociodemographic and WASH data were collected using KoBoToolbox

(questionnaire server) and Open Data Kit (phone application); and data collectors had 4 days

training on how to use the application and fill in the questionnaires. This gives the impression that the study was purely quantitative. I noticed however that under the section on household wash assessment, that, other data collection techniques such as Observation, short interviews and picture taking were used. The PI must mention the items looked out for during the observation and if this was done in every household or in the community as a whole. The issues explored during the short interviews and the respondents of those short interviews must be specified. The PI needs to indicate what pictures were taken and the purpose of the picture taking since nothing is said about the pictures taken in the results section.

It is surprising to see so many young people (26%) and so many women (83.6 %) as household heads. Particularly when over 70% are living together. The PI acknowledges this. Some description of the context of the study area and possible reasons for this unusual findings would make things clearer for the reader.

6. PLOS authors have the option to publish the peer review history of their article (what does this mean?). If published, this will include your full peer review and any attached files.

Reviewer #1: No

Reviewer #2: Yes: Margaret Gyapong

---

## [Author Response · Author response to Decision Letter 0]

20 Apr 2020

Responses on Journal Requirement:

1. Reference style has been checked and files renamed

2. The word slum has been removed

3. Data collection tools have been added through supplementary materials S1 and S2 Appendix

Responses to the Editor:

1. The title of the manuscript has been edited

2. Grammatical and citation errors have been corrected

3. The study design, study period and flow diagram depicting sampling procedure have been added.

4. Rather than sample size determination, the confidence interval for the sample has been added.

5. The conclusion has been summarised

Responses to Reviewer #1:

1. Information has been added to clarify the sampling procedure

2. Data analysis software has been cited

3. Data analysis procedure has been explained in more detail: Odds ratio → Stepwise → Logistic Regression. S3 Appendix shows results of the Bivariate odds ration computations.

4. Grammatical errors have been corrected

Responses to Reviewer#2:

1. Information has been included about the larger study

2. More detail has been given on the research cite selection and sampling procedure (see Figure 1: Sampling procedure)

3. Ethical clearance certificates have been cited

4. Clarification has been made on the sampling method

5. Data collection tools have been added as S1 and S2 Appendix. They have also been clarified in the text. 

6. Peculiar socio-demographic data has been clarified in the text. The definition of household head has also been added to the document.

---

## [Editor Report · Decision Letter 1]

22 Apr 2020

The improved and the unimproved: Factors influencing sanitation and diarrhoea in a peri-urban settlement of Lusaka, Zambia

PONE-D-20-05682R1

Dear Dr. Yamauchi,

We are pleased to inform you that your manuscript has been judged scientifically suitable for publication and will be formally accepted for publication once it complies with all outstanding technical requirements.

With kind regards,

Khin Thet Wai, MBBS, MPH, MA (Population & Family Planning Resear

Academic Editor

PLOS ONE

Additional Editor Comments (optional):

All the comments were responded satisfactorily.
---

## [Editor Report · Acceptance letter]

27 Apr 2020

PONE-D-20-05682R1 

The improved and the unimproved: Factors influencing sanitation and diarrhoea in a peri-urban settlement of Lusaka, Zambia 

Dear Dr. Yamauchi:

I am pleased to inform you that your manuscript has been deemed suitable for publication in PLOS ONE. Congratulations! Your manuscript is now with our production department. 

With kind regards,

on behalf of

Dr. Khin Thet Wai 

Academic Editor

PLOS ONE